# Microbiome-Based Metabolic Therapeutic Approaches in Alcoholic Liver Disease

**DOI:** 10.3390/ijms23158749

**Published:** 2022-08-06

**Authors:** Ji Ye Hyun, Seul Ki Kim, Sang Jun Yoon, Su Been Lee, Jin-Ju Jeong, Haripriya Gupta, Satya Priya Sharma, Ki Kwong Oh, Sung-Min Won, Goo Hyun Kwon, Min Gi Cha, Dong Joon Kim, Raja Ganesan, Ki Tae Suk

**Affiliations:** Institute for Liver and Digestive Disease, College of Medicine, Hallym University, Chuncheon 24253, Korea

**Keywords:** alcohol consumption, gut microbiome, metabolomics, liver injury, fibrosis, steatohepatitis, cirrhosis, hepatocellular carcinoma, liver transplantation

## Abstract

Alcohol consumption is a global healthcare problem. Chronic alcohol consumption generates a wide spectrum of hepatic lesions, the most characteristic of which are steatosis, hepatitis, fibrosis, and cirrhosis. Alcoholic liver diseases (ALD) refer to liver damage and metabolomic changes caused by excessive alcohol intake. ALD present several clinical stages of severity found in liver metabolisms. With increased alcohol consumption, the gut microbiome promotes a leaky gut, metabolic dysfunction, oxidative stress, liver inflammation, and hepatocellular injury. Much attention has focused on ALD, such as alcoholic fatty liver (AFL), alcoholic steatohepatitis (ASH), alcoholic cirrhosis (AC), hepatocellular carcinoma (HCC), a partnership that reflects the metabolomic significance. Here, we report on the global function of inflammation, inhibition, oxidative stress, and reactive oxygen species (ROS) mechanisms in the liver biology framework. In this tutorial review, we hypothetically revisit therapeutic gut microbiota-derived alcoholic oxidative stress, liver inflammation, inflammatory cytokines, and metabolic regulation. We summarize the perspective of microbial therapy of genes, gut microbes, and metabolic role in ALD. The end stage is liver transplantation or death. This review may inspire a summary of the gut microbial genes, critical inflammatory molecules, oxidative stress, and metabolic routes, which will offer future promising therapeutic compounds in ALD.

## 1. Introduction

The past decade has witnessed that the human gut microbiota is a key player in alcoholic liver disease (ALD). The genetic capacity of the gut microbiome evolves the human lifespan. Gut microbes are considered a potential source for new therapeutic biomarker discoveries in ALD. Gut intestine (GI) has a huge collection of various microorganisms (i.e., bacteria, viruses, archaea, and eukaryotes), which are found in human body systems. The majority of microorganisms reside throughout the skin, saliva, oral mucosa, oral cavity, respiratory tract, reproductive systems, and GI [1,2]. GI contains 95 % of the human microbiome (i.e., 100 trillion bacteria and 1000 isolated bacterial species) [3]. Curiously, it has been found that 99% of the human gut microbiota in the GI tract are anaerobic bacteria. The remaining 1% include fungi, protozoa, archaea, and many microorganisms [4,5]. 

The ALD refers to liver disease caused by excessive alcohol consumption, a direct cause of liver disease with a high mortality rate [6]. ALD takes many years to develop. It can start from acute fatty liver to chronic alcoholic cirrhosis (AC), which can then lead to liver cancer (i.e., hepatic carcinoma, HCC) and liver failure. An acute to chronic stage of liver condition, alcohol, or ethanol can contribute to liver cirrhosis and hepatocyte cellular death through an increasing reactive oxygen species (ROS) level [7,8]. 

Gut microbiota contribute to host metabolism via metabolic signaling and microbial pathways. The communication of microbiota with the host-interacting organisms is essential for ALD prevention [9]. The good and bad bacterial flora (e.g., good *Escherichia coli, Lactobacilli, Bifidobacteria,* and bad *Clostridium difficile, Enterococcus faecalis, Lampyllobacter*) are related to liver disease metabolism, obesity, and ALD. Good bacteria are involved in metabolic health maintenance, anti-aging effects, stimulating immunity role in ALD, and act as probiotic bacterium. Bad bacteria are related to pathogens, which could yield metabolites that might impact the metabolic ALD [10]. Strangely, decreased Bacteroides and increased Firmicutes have a wide tendency toward fatty liver formation (i.e., obesity). The phyla *proteobacteria, actinobacteria, fusobacteria,* and *verrucomicrobia* are relatively low in abundance in the gut. Gut microbiome colonization starts at birth and is established in the first three years [11,12]. 

The goal of this review article is to understand only the gut microbiota-derived alcoholic oxidative stress, liver inflammation, inflammatory cytokines, and metabolic regulation in the ALD profile. The molecular mechanisms behind the alcohol effects are not extensively studied in the literature, and therefore, we delineated and limited the current study only to the unexplored area (alterations in microbial gene communities and metabolite profile). The bulk of this review focuses on the microbial community and bioactive microbial products, such as small molecule profiling and disruption in various ALD. ALD exhibits greater healthcare problems and is a microbial disruptor affecting microbiome architecture, metabolic inhibition, and cellular partners. Metabolic functions in the GI tract can allow pathogenic bacteria to colonize, resulting in host infection [8]. Here, we chose to focus on several ALD-based clinical stages, such as AFL, alcoholic liver fibrosis, ASH, AC, and HCC. To address these gaps, the five clinical stages of ALD are discussed in this review manuscript. We hypothetically revisit therapeutic gut microbiota-derived alcoholic oxidative stress, liver inflammation, inflammatory cytokines, and metabolic regulation. The novelty of our manuscript is that acute to chronic ALD is proposed with every stage of metabolic damage.

## 2. Oxidative Stress Formation in ALD 

Oxidative stress is formed by excessive ROS production, which leads to apoptosis and necrosis. Alcohol can enter the liver through the portal vein. Oxidative stress and oxidant stress are pathogenic factors, which result in the production of free radicals, acetaldehyde, and fatty acid ethyl esters that can damage the liver metabolisms [13]. 

ROS is highly involved in ALD. Alcohol-associated hepatocarcinogenesis is oxidative stress, which is secondary to ROS due to alcohol metabolism, inflammation, and expanded iron storage. ROS (i.e., hydrogen peroxide, H_2_O_2_; hydroxyl radical, ºOH; hydroxide, OH^−^, etc.,) can play a significant role in ALD development. Alcohol oxidation via alcohol dehydrogenase (ADH) and the microsomal ethanol-oxidizing system (MEOS) play a significant role in alcoholic metabolism [5]. 

In fact, ROS harms mobile macromolecules that may induce the development of liver carcinogenesis through the formation of lipid peroxides, including 4-hydroxynonenal [14]. The accumulation of ROS causes structural adjustments after damaging the DNA, mainly cell-cycle arrest or apoptosis. The DNA critically impacts genetic functions, including replication and transcription, and cancer proliferation [15]. The accumulation of ROS additionally induces the production of numerous cytokines, the activation of immune cells, angiogenesis, and the metastatic process [16]. Oxidative metabolites, such as acetaldehyde and ROS, produced by alcohol metabolism, can induce epigenetic changes through changes in the metabolism of folate, an essential component of DNA synthesis, and methylation [17,18]. 

Epigenetic regulation involves several chemical modifications, including DNA methylation [19,20]. The genetic polymorphisms in the methylene tetrahydrofolate reductase (*MTHFR*) gene can cause the changes in folate metabolism that have been reported to be associated with the development of AC, HCC [21], and ALD [22]. The oxidation of alcohol through the ADH paths delivers acetaldehyde, which is converted to acetate. Both the oxidation and reduction reaction reduce nicotinamide adenine dinucleotide (NAD) to its reduced form, NAD hydrogen (NADH). 

Alcohol is metabolized into acetaldehyde in the liver, which causes toxicity. Liver cirrhosis is a process in which the normal liver structure is converted into abnormal nodules and eventually into fibrosis [23]. Liver cirrhosis is a chronic form of liver damage, and the causes of cirrhosis include excessive consumption of alcohol, chronic hepatitis C virus infection, and chronic hepatitis B virus infection [24]. Additionally, the symptoms of liver cirrhosis can lead to portal hypertension, enlarged spleen, poor nutrition, ascites, kidney failure, and liver cancer [25].

Fibrogenic mechanisms are initiated and maintained by alcohol metabolism [26]. The metabolism of alcohol in the liver is a highly oxidative event that results in the generation of acetaldehyde [27]. Acetaldehyde is a major toxic metabolite and is one of the major causes mediating the fibrogenic and mutagenic effects of alcohol in the liver [28]. Hepatocytes principally produce acetaldehyde and act on activated hepatic stellate cells (HSC) in a paracrine way, and they directly increase the expression of collagen-I in hepatic stellate cells by activating multiple signaling pathways and transcription factors. Acetaldehyde responds quickly to cellular components, manufacturing adducts, such as malondialdehyde (MDA), 4-hydroxynonenal (4-HNE), and malondialdehyde-acetaldehyde (MAA), which help maintain HSC activation [29].

Drinking excessive amounts of alcohol might upset the equilibrium of intestinal microbes. *Verrucomibrobia* and *Bacteroidetes* grew in alcohol-fed animals. *Firmicutes* largely reduced. *Firmicutes* played a starring role [30]. The particular gut pathogen can cause the GI epithelial cells to produce ROS. Strains of the *Lactobacillus* genus of the GI symbiotic bacteria can cause intestinal phagocytes to produce ROS dependent on NADPH oxidase 1 (NOX1) [31]. Enhanced damage-related molecular patterns (DAMPs) were also produced by increased ROS/RNS, which worsened inflammation. The redox equilibrium of the gut is significantly influenced by gut microorganisms. Alcohol-induced gut microbiota imbalance results in deleterious metabolic expression that affects the liver. 

## 3. Alcohol-Induced Metabolic Inflammation and Cellular Alterations

An inflammasome is a multiprotein oligomer comprising caspase-1, PYCARD, and NALP that promotes the activation and recruitment of inflammatory cells in response to cellular danger signals. The inflammasome initiates procaspase-1, which converts pro-interleukin (IL)-1 to functional IL-1 [32]. Oxidative stress also stimulates the conversion of pro-IL-18 to IL-18, which leads to IFN secretion and natural killer cell activation, as well as the degradation and inactivation of IL-33 [33,34]. Cytosolic caspase activation and recruitment domain (CARD), which recruits caspase-1 to the inflammasome, is also activated by inflammatory stimuli. Inflammasome and IL-1 are activated in ALD patients and rodent models [35]. Recently, multiple inflammasomes had mRNA expression in the liver, implying that inflammasome activation is part of the liver pathophysiology. 

Alcohol metabolites, such as both acetaldehyde and acetate, directly induce an inflammatory response. However, lipopolysaccharide (LPS)-related initiation of proinflammatory cytokines in the Kupper cells/macrophages has activated. Acetaldehyde and acetate exposure of rodent macrophage cells caused the activation of NF-*κ*B signaling and production of TNF-α [36]. This NF-*κ*B activation is, in part, mediated via the downregulation of SIRT1, an NF-*κ*B antagonist; such effect appears to be limited to alcohol metabolites, while alcohol itself has no effect [36]. 

Alcohol-induced intestinal permeability was substantially and negatively linked with *Bifidobacterium* and *Faecalibacterium prausnitzii*, supporting the notion that some bacteria actually strengthen the gut barrier. The anti-inflammatory effects of *Bifidobacterium* and *Faecalibacterium prausnitzii* are well recognized and may be lost due to persistent alcohol use. 

## 4. Alcoholic Liver Diseases 

### 4.1. Alcoholic Fatty Liver and Molecular Networking

Alcoholic fatty liver (AFL) is a build-up of fat inside the liver cells, which helps expand the liver size. Human AFL, liver mortality, and unintended injuries can be elevated among those consuming alcohol over several years [37]. A scar tissue forms when the liver is damaged. This is called liver fibrosis. At the same time, scar tissue replaces healthy tissue. AFL is associated with genetic disorders, lifestyle factors, and social factors. The liver fat concentration can be reduced by heavy alcohol drinking. Worldwide, 90% of chronic liver disease (CLD) and liver failures occur due to alcohol consumption. CLD is a significant public health concern [38]. 

The World Health Organization (WHO) globally estimates that the seventh major cause of death and physical disability is alcoholic disorders [39]. More than 1.32 million people are globally affected by ALD, which is the most common cause of AC in Europe, North America, South America, and central Asia [40]. Over the past two decades, the prevalence of AC has doubled [41]. Alcohol-related cirrhosis and liver cancer account for 1% of all deaths worldwide, and this is expected to increase and create a multilayered burden [42]. 

Human microbiome-derived translational medicinal research has clarified that alcohol consumption could be a global health problem. Alcohol is quickly absorbed from the GI tract and increases the concentration of alcohol in the blood. Alcohol-associated chemicals or small molecules play a significant role in liver diseases. The small molecules are mainly caused by excessive alcohol consumption [43]. Excessive drinking of alcohol causes the liver to become inflamed (swollen), which damages its tissues. ALD patients are affected by hepatitis and cirrhosis. The relationship between ALD and the gut microbiome is often neglected [44]. 

microRNAs (miRNAs) have a central role in protein synthesis. The miRNAs are involved in post-transcriptional modification, miRNAs degradation, and inhibition of miRNAs interactions [45]. In mice, excessive alcohol consumption affected the miRNAs, which led to fatty liver and CLD [46]. CLD has been studied in monozygotic twins, which could explain the miRNAs modification [47]. The genes involved in alcohol metabolism include ADH, acetaldehyde dehydrogenase, and cytochrome P450 2E1 genes that normalize the innate immune response (i.e., Interleukin-1, IL-1; tumor necrosis factor-alpha; TNF-α; and patatin link phospholipase domain containing 3, PNPLA3 genes). Those genes show molecular variances that have been widely considered. The genes of IL-1 and TNF-α related to alcoholic metabolism and AC have been summarized [48,49]. Gut environmental genes and metabolic chemicals may promote the accumulation of fat in the liver. The above genes play an important role in liver metabolic transformation and metabolic dysbiosis from the healthy liver to AFL.

Kupffer cells, stellate cells activation, and macrophages are associated as being against metastatic cancer via CLD. Inflammatory cells, such as monocytes, macrophages, and neutrophils, are derived from the bone marrow, which can migrate to the liver and produce proinflammatory cytokines (i.e., TNF-α, IL-1β, IL-6, monocyte chemoattractant protein-1 (MCP1)) [50,51]. Inflammatory cells in the hepatic sinusoids lead to the activation of stellate cells and the initiation of fibrosis genes (that is, the production of alpha-1 (α-1) smooth muscle actin and collagen-1) [52]. 

As per the publications, studies of fatty liver have shown that alcohol can damage liver cells and activate the apoptotic pathway. The stress on the alcohol-treated endoplasmic reticulum (ER) activates the interferon gene (SIG), which leads to phosphorylation of the interferon regulatory factor 3 (IRF3). Fatty liver cells are recognized by liver immune cells as “danger” signals, leading to proinflammatory reactions [53]. In hepatocytes, alcohol abuse is characterized by an accumulation of fats (especially triglycerides, phospholipids, and cholesterol esters) [54]. 

In AFL, the gut microbiota is essential. *Bacteroidales, Clostridiales, Enterobacteriales* are gaining more attention in AFL. The alteration of gut microbes is brought about by excessive consumption of foods high in calories and alcohol. Due to unbalanced microorganisms, gut dysbiosis develops from chronic alcohol consumption [55]. In particular, alcohol-treated mice exhibited *Bacteroidales* and a few Prevotella OTUs, suggesting the effects of certain strains. *Lachnospiraceae* and *Roseburia hominis* were linked with resistant and FMT-protected mice. Precisely, the downregulated *Lachnospiraceae* and *Roseburia* in AC patients were described [56,57]. The different mechanisms and signaling molecules in mice are represented in Table 1.

### 4.2. Alcoholic Liver Fibrosis in Humans with Alcoholism

The fibrotic tissues are found in presinusoidal and pericentral areas in ALD. Alcoholic steatohepatitis (ASH) is mainly involved in the development of progressive fibrosis [72]. Collagen bands are noticeable and connected to fibrosis expansion. The ASH-based metabolic dysfunction leads to the development of proliferating nodules and cirrhosis of the liver. The cell and molecular metabolic mechanisms of advanced fibrosis, hepatic microcirculatory dysfunction, and vascular occlusion in ALD have not been fully studied [73]. Acetaldehyde (an alcohol metabolite) can directly activate HSC, which are involved in collagen-producing cells in the injured liver. HSC are activated by activated Kupffer cells, damaged hepatocytes, and infiltrating polymorphonuclear leukocytes (PMN) cells. These kinds of cells yield fibrogenic mediators (i.e., growth factors: TGF-β1, PDGF), cytokines (IL-8, TNF-α, leptin, and angiotensin-II), soluble mediators (nitric oxide), and ROS [74]. HSC activate and proliferate under the effect of acute and chronic liver damage [27]. Figure 1 presents the possible mechanisms of gut microbiome-derived metabolic oxidation, inflammation, and inhibition, which could be involved in the development of ALD.

ROS accumulation, mitochondrial damage, ER stress, DNA damage, and protein adducts broadly stimulate the promotion of fibrillation of intracellular signaling pathways in HSCs, including ERK, PI3K/AKT, and c-Jun N-terminal kinases (JNK) [75]. An increase in TIMP-1 and a decrease in the action of metal proteinase promote collagen accumulation. Cells synthesize unprocessed collagen from HSC. These include portal fibroblasts and bone-marrow-derived cells. A novel molecular mechanism in the epithelium-to-leaf transition of hepatocytes may play a key role in alcoholic liver fibrosis [76]. 

The mucosal microbiomes differ in that *Enterococcus, Veillonella, Megasphaera,* and *Burkholderia* are more prominent, and *Roseburia* is far less prevalent [77]. Alcoholics had much lower levels of intestinal bacteria, such as *Enterococci, Bifidobacteria, Eubacterium g23, Oscillibacter,* and *Clostridiales* [78].

### 4.3. Alcoholic Steatohepatitis in Humans with Alcoholism

Patients with severe ASH fail to recover despite abstinence and medical therapy. There was no apparent improvement for the clinical treatment of ASH and AC over three months [79,80]. ASH is a condition caused by regular alcohol use, which results in long-term (chronic) inflammation and altered liver metabolism. Chronic ASH-B, C, and D virus infections are the most common causes of CLD. There are various genotypes of hepatitis-C virus. Heavy alcohol consumption is believed to act synergistically with hepatitis-C in the progression of advanced liver disease [81,82]. Moreover, alcoholic hepatitis, hepatitis-C, and hepatic iron are independent risk factors of higher mortality at 6 months [83]. Hepatitis-C virus is carcinogenic and is associated with the development of different types of malignancies.

Adult human microbiome and microbiota mapping provides a basic outline of metabolic qualities (Figure 2). In ASH, microbial dysbiosis is characterized. Increased *Bifidobacteria, Streptococci,* and *Enterobacteria* in certain species (i.e., *Clostridium leptum* or *F. prausnitzii*) are well-established anti-inflammatory strains. They are found at reduced levels [84]. Another factor in ALD risk is hepatic iron, which could be identified as a predictor of mortality in AC. The molecular mechanism and significance of iron overload in the development of iron-burdened ALD and ASH have not been shown. Now, researchers are focusing on these areas of interest [85].

Acute ASH correlates with disease severity, where it is linked with inflammatory cytokine TNF-α and IL-1β upregulation [50]. The phosphodiesterase inhibitor, pentoxifylline, decreased the TNF-α transcription factor and associated promoter activity. The efficacy of pentoxifylline in ASH has been considered more systematically [86,87]. In particular, in severe ASH, pentoxifylline showed an auxiliary mortality improvement with a low incidence of the contraceptive syndrome [88]. Oxidative stress occurs in ASH. Recent studies have failed to prove the effectiveness of N-acetylcysteine (NAC) treatment in ASH [89,90,91,92]. Interestingly, a combination therapy with corticosteroids plus NAC increased the 1-month survival rate among patients with severe acute ASH, but the 6-month survival rate did not improve [92]. The ineffectiveness of anabolic steroids in ASH has been formerly revised [93]. Propylthiouracil (PTU) counters alcohol-induced hypermetabolism and suppresses oxidative stress metabolism [94,95,96]; however, a recent review of PTU exhibited that it is ineffective in ALD [97,98]. 

Ascites and jaundice are the main symptoms, and a significant number of patients have hepatic encephalopathy [99]. Generally, as per the current guidelines, liver transplantation is not advised for patients with ASH [100]. Liver biopsy shows progressive degeneration, focal hepatocyte necrosis, and neutrophilic infiltration [101]. Despite the good outcomes in various publications, liver transplantation as a treatment for ASH remains controversial, and there is currently an organ shortage. Hence, in most cases, liver transplantation is not recommended as the treatment option for ASH.

ASH is formed by hepatocellular damage and parenchymal inflammation and is a prerequisite for the development of fibrosis and cirrhosis. An episode of ASH may cause severe liver damage, increase resistance during blood circulation, and may be associated with a poor prognosis [102]. Many molecular mechanisms can contribute to the development of ASH. Acetaldehyde has a toxic effect. It binds to proteins and DNA to form self-antigens, resulting in fundamental changes and protein adducts that activate the immune system [103,104]. Mitochondrial damage and impaired glutathione functions develop, leading to oxidative stress and apoptosis [105]. The generation of ROS accompanying the formation of DNA damage and lipid peroxidation is initiated [106]. Major sources of ROS include the CYP2E1-dependent MEOS respiratory chain mitochondrial transport system, NADH-based cytochrome reductase, and xanthine oxidase, which are significantly correlated with ASH metabolism [107,108]. 

Chronic alcohol consumption has remarkably increased the CYP2E1 gene, which could metabolize ethanol to acetaldehyde and increased ROS and hydroxyl-ethyl radicals [109]. Alcohol metabolites and ROS interfere with signaling pathways, such as natural killer (NK) cells kβ, STAT-JAK, and JNK, in hepatocytes to induce local stimulation of inflammatory processes, such as TNF-α, CXC chemokines (e.g., IL-8), and osteopontin [110]. Alcohol intake may cause elevations of the gut microbial flora of the large intestine and serum lipopolysaccharides, which cause changes to intestinal permeability [111,112]. This induces inflammatory action by CD14/TLR4 activity in Kupffer cells [113]. Consequences of the inflammatory environment in alcoholic liver include PMN infiltration, ROS formation, and hepatic impairment. The reduction in the metabolic pathway to the ubiquitin proteasome leads to hepatocyte damage and the presence of aggregated sites of cytokeratin (i.e., Mallory–Denk bodies) in the liver [114].

Patients with ASH had higher concentrations of *Acidaminococcus, Escherichia* spp., and *Bacteroides* spp., which are linked to insulin resistance, and lower levels of *Lachnospiraceae* and *Ruminococcaceae*, which are responsible for butyrate production. These findings are consistent with those of a prior study, which found that *Bacteroides vulgatus* was one of the major species contributing to insulin resistance and circulating BCAA levels in humans [115]. 

The host gene Muc2 expression is primarily limited to the intestine and does not occur in the liver or in inflammatory cells. The Muc2 gene may cause ASH death. Probiotic *Lactobacillus* helped mice whose ability to restore ASH was impeded [116,117]. *Bacillus* and *Veillonella* were increased in the feces of patients with severe alcoholic hepatitis relative to healthy subjects [118]. 

Patients with ASH frequently have bacterial infections found. A crucial mechanistic role in ASH is played by the movement of bacteria and the bacterial proteins they produce throughout the GI tract. *P. gingivalis* is a significant periodontal pathogen that causes chronic periodontitis and can potentially have an impact on distant organs, such as the liver [119]. *P. gingivalis* might represent a brand-new ASH risk factor.

### 4.4. Alcoholic Cirrhosis in Humans with Alcoholism

AC is a result of severe damage to the liver cells. In this advanced stage of cirrhosis, the liver becomes stiff, swollen, and barely able to function. Hepatitis-B virus (HBV) and hepatitis-C virus (HCV) are major risk factors for AC development. HCV infection is one of the leading end-stage liver cirrhosis conditions that requires liver transplantation. 

Alcohol increases the activity of cytochrome P450 2E1 (CYP2E1), which can metabolize alcohol and produce ROS [120]. ROS forms through CYP2E1-dependent oxidative ethanol metabolism that can involve collagen production in HSC cocultured with hepatocytes [26]. Ethanol metabolism induces oxidative stress and the secretion of inflammatory cytokines that can activate HSC. The triggered HSC move to the area of liver injury and secrete many ECMs, which act as the main events triggering the process of liver fibrogenesis [121]. In this study, HepG2 cells that do (E47 cells) or do not (C34 cells) have CYP2E1 with HSC were used to evaluate the potential fibrogenic effects of CYP2E1-dependent creation of ROS. Both intracellular and extracellular H_2_O_2_ and lipid peroxidation were increased in HSC-incubated E47 cells. This suggests that ROS is triggered by CYP2E1 metabolism. The HepG2 cells may spread and enter HSC after modulation of the collagen type-I protein. However, collagen type-I protein increased in AC [122]. We summarize the understanding that alcohol induces metabolic liver damage through the gut–liver axis (Figure 2).

Acetaldehyde binds to proteins to produce byproducts [123]. Malondialdehyde (MAA) and acetaldehyde react with proteins in vivo during liver ethanol metabolism to produce an MAA adduct, a hybrid protein. A study found that the absorption of chronic ethanol produces high levels of MAA-added proteins, which are associated with liver damage in humans with ALD [124]. MDA is also produced during lipid peroxidation in hepatocytes by aldehyde dehydrogenase. Both MDA and acetaldehyde can produce stable adducts, and while ethanol is oxidized in the liver, MDA and acetaldehyde can coexist at similar concentrations. In addition, the concentration of bio proteins increases, dependent on the coexistence of malonaldehyde and acetaldehyde, and MDA increases when the protein forms a structurally stable bond with acetaldehyde [125]. The adduct of ethanol metabolism and the MDA product of lipid peroxidation appear to progress during liver necrosis and liver fibrosis [126]. This includes oxidative tissue damage to the liver during the process of metabolizing excessive absorption of ethanol by the liver. After 12 months of ethanol consumption, osteoblastic fibrosis is widely developed. MDA additives caused by lipid peroxidation of chronic alcohol in the liver lead to liver necrosis and fibrosis [127]. 

AC is a major public health problem and is caused by excessive alcohol intake [128]. The molecular and analytical link in AC needs to be understood [129]. Liver cirrhosis, an irreversible liver disease, is accompanied by ascites and jaundice, and it has poor prognosis [130]. Studies have shown that one-third of patients die from liver cirrhosis during hospitalization, and the remaining 60 % only live for a short time [131]. As the most serious consequence of alcohol abuse, we revisited several metabolic chemical reactions that are necessary to reduce the negative consequences of AC [132]. 

In AC patients, there was an increase in *Prevotellaceae*. On the other hand, patients with ALD had lower levels of *Firmicutes* and *Bacteroidaceae* [57]. The *Actinobacteria* and *Proteobacteria* of the Gram-negative bacteria were raised as a result of heavy alcohol use. The *Gammaproteobacteria* class, which includes the *Enterobacteriaceae* and *Pasteurellaceae*, was primarily responsible for the enrichment of *Proteobacteria*. Patients with AC had 27 times more *Enterobacteriaceae* than healthy controls in their feces. In individuals with cirrhosis, *Enterobacteriaceae* were the most prevalent liver-translocating bacteria [133]. 

### 4.5. Hepatocellular Carcinoma in Humans with Alcoholism

Hepatocellular carcinoma (HCC) is the fifth and seventh most frequent cancer in men and women, respectively [134,135]. The incidence of HCC has been frequently amplified worldwide. Among patients with AC, 1–2 % develop HCC [136]. The CLD significantly increases the risk of HCC. HCC is the most common hepatic malignancy. The use of metabolomics for HCC prevention is still at an early stage, although it has many promising findings. There are few reports or studies about mortality prediction using HCC biomarkers.

In the liver cellular milieu, Kupffer cells and bone-marrow-derived macrophages recognize small sequences of molecules, formally called pathogen-associated molecular patterns (PAMPs), from endotoxins from the main circulation via TLR-4 [137]. The upregulation of TLR-4 promotes the binding of its ligand of myeloid differentiation primary response protein 88 (MyD88). This can induce the mitogen-activated protein kinase (MAPKs), p38, and (JNK). The inhibition of nuclear factor kappa-B (NK-*κ*B), mitogen-activated protein kinase (MAPK), p38, and the NK-*κ*B signaling is promoted by TLR-4 modification. These effects favor the release of TNF-*α*, IFN-*γ*, prostaglandin-2, chemokine C-C motif ligands, IL-1*α*, IL-1*β*, IL-6, ROS, and nitrates that can tolerate liver inflammation [138,139]. 

From the various carcinogenic effects, NF-*κ*β can induce the expression of antiapoptotic genes (TRAF-1 and TRAF-2) [8,140]. TNF-*α* deregulates the tight junctions (TJ) and induces an intestinal barrier disorder. Surprisingly, higher levels of TNF-*α* and IL-6 were found in alcohol-dependent focused duodenal biopsies, which could be established in in vivo studies [141]. According to publications, alcohol-dependent DSM-IV criteria, TNF-*α*, IL-6, and IL-10 showed important alterations that were considered candidate therapeutic biomarkers [142]. 

Moreover, IL-37 requires anti-inflammatory receptors through IL-18R*α* and IL-1R8 for transportation from the extracellular space to the cytoplasm. IL-37 expression is significantly lower than that in non-AFL patients [143]. According to in vivo analysis, IL-37 expression was downregulated in wild-type mice after ethanol administration [143]. This change in IL-37 expression plays important roles in HCC. IL-37 is correlated with tumor size and is linked to disease-free survival and quantity by inducing tumor-infiltrating CD571 natural killer cells [144]. Molecular variations in HCC and their biological interpretation are fully documented through genomics. Clinical therapeutic targets (i.e., MyD88, IRAK4, IRAK1, TRAF6, IKK*β*) and the discovery of anti-inflammation pathways (i.e., AMPK, STAT3, STAT6, MER, PTEN) have been summarized in liver cancer [145,146]. 

In liver cell metabolism, TLR-4 expression has been found in hepatic stellate cells (HSC), hepatocytes, and endothelial cells [147]. In HSC, an upregulation of hepatocyte epiregulin has been found [148], which stimulates epidermal growth factor protein, which has mitogenic properties in hepatocytes [149]. The antiapoptotic properties of NF-*κβ* promote hepatocarcinogenesis development. TLR-4 deficiency and intestinal stabilization analysis in knockout mice have been studied in steatosis, oxidative stress, and inflammation, with a resulting decrease in HCC risk. The risk of liver injury was increased by the lack of innate immunity triggered by TLR-4 suppression [150]. Chronic alcohol consumption is linked to immunosuppression with reduced CD8 T cells, which play an important role in antitumor effects [151]. In Table 2, a summary of gut microbiota-derived mechanistic candidate publications is presented with various targets for human liver treatment responses in ALD. Finally, abstinence is most important to prevent liver injury and is beneficial at all stages of liver diseases. We suggest that multidisciplinary approaches are required to treat alcohol disorders. 

All patients with cirrhosis also had lower levels of *Akkermansia* and higher levels of *Enterobacteriaceae*, whereas the HCC subset had higher levels of *Bacteroides* and *Ruminococcus* with lower levels of *Bifidobacterium*. When a correlation network was built, these patterns were discovered to be constant within the increased inflammation. In contrast to *Bacteroides*, which were linked to higher levels of the proinflammatory cytokines IL8 and IL13, this network revealed an inverse association between *Akkermansia* and fecal calprotectin [162,163]. ALD is a major indication of liver transplantation worldwide.

## 5. Probiotics and Antioxidant Activity in ALD

As per a new study, pre-/probiotics are living microorganisms that play a beneficial role. Probiotics can be obtained in a variety of ways, including diet and human gut microbiota. The TJ proteins occludin and claudin-3 were frequently expressed by the probiotic *Akkermansia muciniphila*, which improved the liver damage caused by alcohol. *Akkermansia muciniphila* plays a more promising probiotic role [164,165]. Alcohol consumption quantitively decreased *Bacteroides* spp. and *Firmicutes.* Similarly, *Proteobacteria* and *Actinobacteria* were increased by alcohol [166]. *Lactobacillus spp.* has been shown to improve various liver diseases. *Lactobacillus* and *Bifidobacterium lactis* were used together to treat functional bowel disorders and ALD. In liver and intestine, probiotics were involved in modulating the gut microbiota and immune response, decreasing inflammatory cytokine and ROS expression. Probiotics play a significant role in reducing the fat accumulation in liver while increasing the fatty acid β-oxidation [167]. 

Nutritional care based on prebiotics and probiotics is an important part of ALD treatment. Probiotics were shown to be effective in reducing or preventing some ALD. Probiotics are based on quantitative changes in bacterial overgrowth in the intestine, which is common in ALD patients. These probiotics of *Akkermansia muciniphila* and *Lactobacillus* spp. could help improve the ALD survival rate. Recent studies investigating the use of pre-, pro-, and symbiotics in ALD and cirrhosis have found that they improve clinical and biochemical markers of liver disease [168].

## 6. Microbiome-Wide Dynamic Microbial Proof-of-Concept Clinical Validation

The relationship between the gut microbiome and liver diseases (i.e., dysbiosis, fibrosis, AFL, ASH, AC, and HCC) is more complex than the involvement of the microbiota in other diseases. Norfloxacin and rifaximin are used to increase the survival rate of patients with cirrhosis and HCC [169,170,171,172,173,174]. These drug treatments had better efficacy. These drugs are associated with severe liver disease (i.e., hepatic encephalopathy, portal hypertension, and spontaneous bacterial peritonitis) [175,176]. Probiotic drug-based gut microbes with modulating effects are being developed in order to restore bacterial diversity. Prebiotics have the effect of alleviating alcoholic liver injury [177]. At present, many pharmaceutical companies have made profits with strong microbial mixtures sold in both pharmacies and superstores. This kind of bacterial microbial compound has not been proven effective in cancer environments. However, bacterial microbial compounds have shown good efficacy in GI conditions, such as *Clostridium difficile* infection (CDI), where they can distribute fecal microbiota by transplantation and improve the anticancer immune response by preventing cancer progression [178]. CDI is notable for its increased virulence in ALD. However, these kinds of treatments are not included in the protocol of HCC treatment due to a lack of standardization. New clinical analysis is focusing on gut microbiota-based liver dysbiosis to increase therapeutic options. It was also seen in various studies that HCC served as an advanced stage of liver diseases. Nowadays, a variety of anti-inflammatory and anti-cytokine therapies are being used to treat ALD (Figure 3). However, these treatments were not very effective, and 15 % of patients continued to experience a worsening of liver function. Liver transplantation is the only option for people with end-stage or advanced ALD [179]. Accurate non-invasive metabolomic techniques to replace liver biopsy are a gold standard diagnostic tool for ALD. Death could occur due to hepatic failure induced by ALD. Finally, an emerging technology, such as miRNA analysis and artificial-intelligence-based methods to examine the metabolomic, genomic, and overall functional profile, are particularly promising.

## 7. Summary and Future Outlook

According to the present study, we live with a huge number of microbes in our gut, ranging from metabolic inhibition, inflammation, and various metabolic stresses. There are no questions about the fact that microbiota-based metabolomic signatures in ALD have been reported. ALD has rapidly become a global health problem. It is urgent to provide new therapeutic microbiota-associated genetic factors, proteins, and metabolites, especially for late-stage ALD. The therapeutic biomarkers of IL-1α, IL-1β, IL-6, IL-37, TNF-α, NK-kB, TGF-β1, CD14, CD571, and metabolites were summarized in various ALDs. Metagenomic comparisons predict genomic functions. Metabolomics provides an expansive biochemical profile of individual microbial strains. The clinical application of gut microbiota-based metabolomic signature flows, small molecule databases, and novel techniques specific to metabolomics profiling are useful and attractive platforms. 

The probiotics of *Akkermansia muciniphila* and *Lactobacillus* spp. play a significant role in ALD survival rate. Probiotics provide a promising new approach for improving ALD. In food science research, probiotics explore new paths of ALD survival rate extension. Probiotics provide promising new approaches to improve ALD.

The idea is to examine whether physical exercise changes the microbiota, so that the accumulation of fat in the liver decreases. The gut and the liver are in constant interaction with one another (for example, through metabolic compounds produced by gut bacteria). Some chemicals found in the gut–liver microenvironment are known as metabolic disruptors that affect glucose and fat metabolism. We hope that the microbiota-based genetic and metabolic profile role of ALD will provide input to the scientific community discussion. As a first step, we plan to outline various gut microbiota, phenotypic variations, and metabolite information, extending to biomarker discovery, which may be helpful in underlining phenotypic mechanisms. This review can be used as a direct reference and to suggest new molecular targets for therapeutic interventions in ALD.

## Figures and Tables

**Figure 1 ijms-23-08749-f001:**
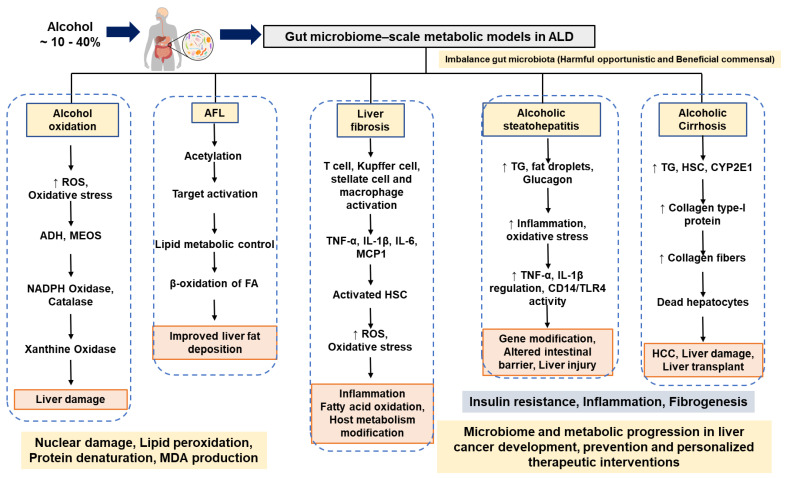
Summary of possible mechanisms and discussions of gut microbiome-derived metabolic oxidation, inflammation, and inhibition, which could be involved in the development of ALD. The possible metabolic influence of liver dysbiosis on ALD. Abbreviations: ALD, alcoholic liver disease; AFL, alcoholic fatty liver; TG, triglyceride; TNF-α, Tumor necrosis factor-α; IL, interleukin; ROS, reactive oxygen species; HS, hepatic stellate cell; NADPH, Nicotinamide adenine dinucleotide phosphate; ADH, alcohol dehydrogenase; TLR, Toll-like receptor.

**Figure 2 ijms-23-08749-f002:**
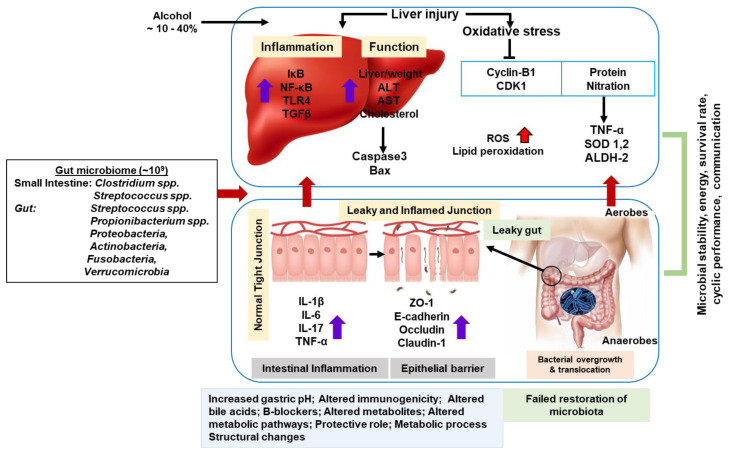
Proposed models for understanding the alcohol-induced liver metabolic damages through gut–liver axis.

**Figure 3 ijms-23-08749-f003:**
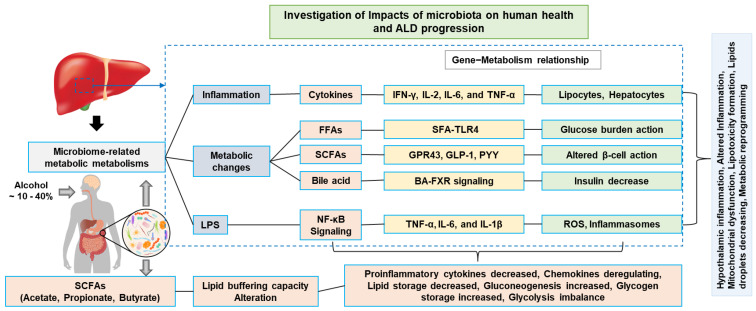
Proposed pathogenic mechanisms of ALD. An alcohol-induced liver metabolic intracellular imbalance through gut–liver axis. LPS, lipopolysaccharide; FFAs, free fatty acids; SCFAs, short-chain fatty acids.

**Table 1 ijms-23-08749-t001:** Robust predictions of the recent genomics and metabolomics in alcoholic animal models.

Animals		Exposures	Main Results	Ref.
Mice	C57BL/6J male (6–8 weeks old)	chronic 5% ethanol diet for 11, 22, and 33 days	(↑) AST, ALT, amount of G-MDSC.	[58]
	C57BL/6 mice	alcohol diet for 8 weeks	(↑) proportions of CA, total MCA, DCA in ileum unconjugated and total bile acid concentrations in the plasma hepatic Cyp7a1 protein expression, hepatic IL-1B, TNF protein. (↓) proportions of TCA and TDCA in ileum.	[59]
	C57BL/6J female (7–8 weeks old)	5% ethanol for 6 weeks in Lieber–DeCarli liquid diet	(↑) liver mRNA expression level of TNF-α, IL-6, Cc2, Ccr5 liver protein level of TNF-α, IL-1β, IL-6, CD14 serum protein level.	[60]
Rats	Male Wistar rats	Non-stop ethanol supply for 3 weeks. Gut sterilization with polymyxin B and neomycin	(↓) Plasma endotoxin levels (80–90 pg/mL → <25 pg/mL), average hepatic pathological score in ethanol-fed and antibiotic-treated rats. Antibiotics prevented elevated aspartate aminotransferase levels and hepatic surface hypoxia.	[61]
Mice	Alcohol-fed NS5ATg mice	Lieber–DeCarli diet containing 3.5% ethanol or isocaloric dextrin for long-term alcohol feeding, repetitive LPS injection	(↑) Ethanol-induced endotoxemia, liver injury, and tumorigenesis after Toll-like receptor (TLR)-4 induction through hepatocyte-specific transgenic expression of the HCV non-structural protein NS5A.	[62]
Mice	60 male Kunming mice (18–22 g) (6–8 weeks old)	alcohol gavage for 2–13 days	(↑) AST, ALT, TG, Hepatic MDA, ADH, mRNA, and protein expression of Cyp2e1, CAT. (↓)Major endogenous antioxidant enzymes (SOD and GSH-Px) mRNA expression of Nrf2, NQO-1, ADH.	[63]
Rats	Male Wistar rats (170–180 g)	chronic ethanol feeding	(↑) ROS production by LPS in Kupffer cells isolated from ethanol-fed mice. ROS production in Kupffer cells by LPS stimulation were increased NADPH oxidase dependently. ERK1/2 contributed to the increase in TNF-α production in Kupffer cells by LPS stimulation.	[64]
Mice	C57BL/6 male	EtOH-containing diets (35% of total calories, AF) ad libitum for 4 weeks	(↑) Saturated fatty acid levels. PLS-DA performed for liver and fecal samples. Mouse liver damage can be improved. (↑) intestinal; (↓) hepatic fatty acids; (↑) amino acid concentration.	[65]
Mice	C57BL/6 male (5–6 weeks old)	EtOH-containing Lieber– DeCarli liquid diet or an isocaloric control diet	(↑) ALT and AST. PCA, OPLS-DA, volcano maps, and correlation coefficient analyzed.	[66]
Mice	C57BL/6 male (8 weeks old)	Intermittent hypoxia exposure	PCA, OPLS-DA, and volcano maps, heatmaps analyzed. (↑) N1-(5-Phospho-D-ribosyl)-AMP, stearidonic acid, adenine, arachidonic acid (peroxide-free), ergothioneine, betaine, cyclohexylamine, GSH, GSH disulfide.	[67]
Mice	Kunming mice (7 weeks old)	10% lard, 20% sucrose, 2.5% cholesterol, and 0.5% sodium cholate	(↑) Taurochenodeoxycholic acid, taurine, chenodeoxycholic acid, (4Z,7Z,10Z,13Z,16Z,19Z)-4,7,10,13,16, 19-docosahexaenoic acid, oleic acid, alpha-linolenic acid. Enrichment analysis.	[68]
Rats	Male Sprague Dawley rats (1 year old, 180–200g)	CCl_4_ (1mL/kg 40% CCl_4_, diluted in olive oil); twice a week for eight weeks	H&E and Masson’s trichrome staining, PLS-DA, heatmap, ROC curve analysis. (↑) L-tryptophan, L-valine, cholesterol, glycocholate, methylmalonic acid.	[69]
Mice	BALB/c mice (8 weeks old)	*E. granulosus* infection	25mg of hepatic and fecal samples were analyzed. PCA, OPLS-DA, and volcano maps, heatmaps analyzed. (↑) 2-ethyl-2-hydroxybutyric acid, 2-hydroxyvaleric acid, cytidine 2’,3’-cyclic phosphate, sodium citrate, carboxytolbutamide, methylselenopyruvate. (↓) Pyronaridine, Bis(4-nitrophenyl) phosphate, Inosinic acid, 5-Phosphoribosyl-4-carboxy-5- aminoimidazole, tolclofos-methy, maduropeptin chromophore.	[70]
Mice	Fat-1 transgenic mice (10–12 weeks old)	EtOH diet	(↑) neutrophil accumulation, Pai-1 expression in wild-type mice. (↓) neutrophil accumulation, pai-1 expression, KC M1 abundance in fat-1 mice. Flow cytometry analysis of hepatic immune cells.	[71]

Notes and abbreviations: ↑, and ↓ show an increase and decrease in the condition. EtOH, ethanol; AST, aspartate transaminase; ALT, alanine transaminase; G-MDSC, granulocyte-like myeloid-derived suppressor cells; ADH, alcohol dehydrogenase; ALD, Alcoholic liver disease; IL, interleukin; LBP, Lipopolysaccharide binding protein; LPS, Lipopolysaccharide; NADPH, Nicotinamide adenine dinucleotide phosphate; ROS, reactive oxygen species; TLR-4, Toll-like receptor-4; TNF-α, Tumor necrosis factor-α; AHB, asymptomatic hepatitis B virus infection; CHB, chronic hepatitis B; CHC, chronic hepatitis C; CIR, cirrhosis type C; HCV, Hepatitis C virus; HCC, hepatocellular carcinoma; GSH, Glutathione; NAFLD, non-alcoholic fatty liver disease; PLS-DA, Partial least-squares discriminant analysis; OPLS-DA, Orthogonal PLS-DA; *E. granulosus,* Echinococcus granulosus.

**Table 2 ijms-23-08749-t002:** Robust predictions of the recent genomic and metabolic properties in humans with ALD.

Animals		Exposures	Main results	Ref.
Human	14 alcoholic patients	chronic alcohol intake	(↑) Plasma endotoxin levels and serum IL-6 and IL-8 levels of patients compared to healthy subjects. Serum LBP was positively correlated with white blood cell and neutrophil counts as an indicator of an inflammatory response.	[152]
	recombinant HepG2 ^ADH1/CYP2E1^ cells	100 mM ethanol for 6, 24, 48, 72, 96, and 110 h	(↓) CYP1A2, CYP2B6, CYP2C9, CYP2E1, and CYP3A4 expression. (↓) AGO1 knockdown, HNF4A RNA levels.	[153]
	severe AH (n = 161), and HC (n = 28)	chronic alcohol intake	(↑) level of sST2 was increased in SAH, higher levels of 3-HM in patients compared with controls, expression at baseline of GRK2 in circulating PMNs. (↓) expression of the chemokine receptor CXCR2 on the surface of circulating PMNs.	[154]
Human	51 alcoholic patients	consumed excessive alcohol, tobacco smoking	(↑) CYP2E1 activity, oxidative stress. (↑) chlorzoxazone oxidation.	[109]
	10 liver samples of AC	chronic alcohol intake	(↑) increased CCL2, CCL3, CCL4, CCL5, CCL8, CCL5 mRNA expression in AC liver, increased MØ infiltration.	[60]
	healthy control (n = 33), alcoholic liver cirrhosis (n = 23)	chronic alcohol intake	(↑) tumor volume and tumor maximum diameter expression of BCL-xl, CCL2, IL-4, IL-10, TIMP1, col1a1, and PCNA the frequency and number of macrophages in the liver hepatic CD206 expression. M2-associated protumor genes in the liver.	[151]
Human	53 cirrhosis cohort patients	alcohol intake, 1 yr follow-up, underwent liver transplantation	Small intestinal bacterial overgrowth was seen in 59% of patients with cirrhosis and was significantly related to systemic endotoxemia.	[155]
	AH patients (n = 6); HC persons (n = 6)	Ethanol consumption of at least 80 g/day	(↑) NF-*k*B activity in the monocytes of six patients with AH as compared with normal subjects. (↑) NF-*k*B activity, TNF-*α* RNA expression, and TNF-*α* release by endotoxin in AH patients.	[156]
Human	HCC, Late intrahepatic recurrence (n = 18); Early intrahepatic recurrence (n = 22)	HCC patients	(↓) Plasma specimens, tryptophan, cholesterol glucuronide, LysoPC (20:5), LysoPC (22:6). ROC curves based on methionine, GCDCA, and cholesterol sulfate was selected. AUC equal to 0.95.	[157]
Human	46 patients	HCV-related HCC patients	PCA and PLS-DA score-plot has found. ROC curve analyzed for N-acetyl-lysine, L-glutamine, L-aspartate, and L-proline. Heatmap presenting the hierarchical clustering analysis.	[158]
Human	248 serum samples	AHB, CHB, CHC with many types of liver disease	Heatmap analysis, *γ*-glutamyl peptides mechanism, GSH oxidation and reduction. (↓) GSH level.	[159]
Human	52 serum samples	HCV, HCC patients	Serum sample analysis, 73 metabolites detected, Sensitivity of 97%, specificity of 95%, and an AUROC of 0.98 found. Sixteen metabolites were significantly altered.	[159]
Human	559 patients	NAFLD patients	AUROC of 0.92, sensitivity of 73%, and specificity of 94%.	[160]
Human	117 patients	HCV (n = 67), HBV (n = 50 patients)	OPLS-DA analysis, metabolites and their pathway analysis, Fold-change analysis.	[161]

Notes and abbreviations: ↑ and ↓ show an increase and decrease in the condition. ALD, Alcoholic liver disease; IL, interleukin; LBP, Lipopolysaccharide binding protein; LPS, Lipopolysaccharide; NADPH, Nicotinamide adenine dinucleotide phosphate; ROS, reactive oxygen species; TLR-4, Toll-like receptor-4; TNF-α, Tumor necrosis factor-α; AHB, asymptomatic hepatitis B virus infection; CHB, chronic hepatitis B; CHC, chronic hepatitis C; CIR, cirrhosis type C; HCV, Hepatitis C virus; HCC, hepatocellular carcinoma; GSH, Glutathione; NAFLD, non-alcoholic fatty liver disease; OPLS-DA, Orthogonal Projections to Latent Structures Discriminant Analysis.

## Data Availability

Data are contained within the article.

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
