# Peer review of "Microbiome-Based Metabolic Therapeutic Approaches in Alcoholic Liver Disease"

_ijms, 2022, doi:10.3390/ijms23158749_

Round 1
Reviewer 1 Report
The authors made a Review regarding Microbiome-based Metabolic Therapeutic Approaches in Alcoholic Liver Disease. However, the topic is somehow interesting, more data must be added. Only 14.5 pages for a review, in the MDPI format (2/3 of a page is occupied with text) seems not complete.
Aim of the study is missing. Please make the aim of the study a separate, last paragraph of Introduction section (to make it easier visible for those interested in the topic), highlighting better some aspects by responding to the following questions: Which is the novelty of your study or the special aspects it brings to the field? What makes different your study from others in the same/similar topic, already published?
L76. Alcohol not the same with ethanol. Please revise.
Please describe the role of supplementation with prebiotics in patients with alcohol related liver disease. Make a separate chapter where you can do a table indicating specific studies where probiotics were used in alcoholic liver disease. Also describe the influence of probiotic supplementation on total antioxidant capacity of the blood and the role of prevention in alcohol hepatitis.
Please detail if certain bacteria have superior beneficial role compared to other species. I suggest checking and referring to https://doi.org/10.3390/microorganisms9030618
Author Response
First of all, we would like to thank the Reviewer-1 and 2 for his/her comments, which helped us to improve this manuscript.
# Reviewer 1:
Comments and Suggestions for Authors
The authors made a Review regarding Microbiome-based Metabolic Therapeutic Approaches in Alcoholic Liver Disease. However, the topic is somehow interesting, more data must be added. Only 14.5 pages for a review, in the MDPI format (2/3 of a page is occupied with text) seems not complete.
Response: We are grateful for the reviewer’s valuable comments. Much thanks for your scientific commentaries and appreciation of our review works. As per your concerns, we added the following details in manuscript and revised whole manuscript as reviewer pointed.
According to IJMS guidelines, there is no page limit for review manuscript. Also, this manuscript is covered in bulk section of ALD. Our chosen topics are very big. We feel, this is mandatory in ALD scientific community.
Aim of the study is missing. Please make the aim of the study a separate, last paragraph of Introduction section (to make it easier visible for those interested in the topic), highlighting better some aspects by responding to the following questions: Which is the novelty of your study or the special aspects it brings to the field? What makes different your study from others in the same/similar topic, already published?
Response: As per your comments and suggestions, we edited the last paragraph in introduction section.
The goal of this review article is to understand only the gut microbiota-derived alcoholic oxidative stress, liver inflammation, inflammatory cytokines, and metabolic regulation in ALD profile. The molecular mechanisms behind the alcohol effects are not extensively studied in the literature and therefore, we had delineated and limited only to the unexplored area (alterations in microbial gene communities and metabolite profile). The bulk of this review focuses on the microbial community and bioactive microbial products such as small molecule profiling and disruption in various ALD. ALD exhibits greater healthcare problems and is a microbial disruptor affecting microbiome architecture, metabolic inhibition, and cellular partners. Here, we have chosen to focus on ALD-based several clinical stages, such as alcoholic fatty liver (AFL), alcoholic liver fibrosis, alcoholic steatohepatitis (ASH), alcoholic cirrhosis (AC), and hepatocellular carcinoma (HCC). To address these gaps, the five clinical stages of ALD have been discussed in this review manuscript. We hypothetically revisited therapeutic gut microbiota-derived alcoholic oxidative stress, liver inflammation, inflammatory cytokines, and metabolic regulation. The novelty of our manuscript is that acute to chronic ALD has been proposed with every stage of metabolic damage.
L76. Alcohol not the same with ethanol. Please revise.
Response: We agree with the reviewer’s comment. It is now corrected in the revised manuscript.
Please describe the role of supplementation with prebiotics in patients with alcohol related liver disease. Make a separate chapter where you can do a table indicating specific studies where probiotics were used in alcoholic liver disease. Also describe the influence of probiotic supplementation on total antioxidant capacity of the blood and the role of prevention in alcohol hepatitis. Please detail if certain bacteria have superior beneficial role compared to other species. I suggest checking and referring to https://doi.org/10.3390/microorganisms9030618
Response: As reviewer pointed out, we added the new chapter (numbered in section 5. Probiotics and antioxidant activity in ALD). We referred the specific article and cited in our manuscript. Due to size and length of article, we added important point of pre- and probiotics. In our current working manuscript, we will discuss more deeply about pre-/ probiotic role in liver disease at cellular microenvironmental level.
Reviewer 2 Report
This paper purports to summarize microbiome-based metabolic approaches for ALD. However, in almost every section, only the pathologies associated with each stage of ALD and the various known mechanisms are summarized, but with little or no evidence of how the gut microbiome mediates these changes. Their figures have included "gut microbiome" but little additional evidence of the role in each stage. Only at the end of the paper, do they mention that probiotics are being developed, but no further elaboration is provided. After reading this paper, one comes away with little in the way of how leveraging the microbiome can be used to treat ALD.
Author Response
# Reviewer 2:
Comments and Suggestions for Authors
This paper purports to summarize microbiome-based metabolic approaches for ALD.
Response: We value the reviewers’ comments. Thank you for your valuable positive opinion about our manuscript.
However, in almost every section, only the pathologies associated with each stage of ALD and the various known mechanisms are summarized, but with little or no evidence of how the gut microbiome mediates these changes.
Response: Thank you very much for your comments. Microbiome and liver function includes multifold interactions in homeostasis and disease. Host microbiome-metabolite crosstalk is very big part. However, we highlight the host microbiome-metabolite-based known mechanisms were summarized in AFL, ASH, AC, and HCC.
Our approach is one of the first to systematically organize potentially important, microbial genes- metabolites crosstalk at scale, especially among non-housekeeping processes that may only manifest in a gut microbiome environment. The brief explanation and illustration of ALD strategies are summarized. It will briefly be describing the fundamental molecular approaches from initial stage AFL to late-stage HCC.
Their figures have included "gut microbiome" but little additional evidence of the role in each stage. Only at the end of the paper, do they mention that probiotics are being developed, but no further elaboration is provided. After reading this paper, one comes away with little in the way of how leveraging the microbiome can be used to treat ALD.
Response: We agree with the reviewer’s comment and apologize for causing shortage of probiotics evidence. We added new chapter about probiotics and antioxidant activity in ALD (Section number 5). The significant gut microbiome is shown in Figure 2. The relevant publications in ALD are cited. Several microbiomes, genes, metabolites regulation are marked in this bulk review manuscript. Microbiome arenas play big role to cause ALD. We hope that this manuscript will open up new avenues for gut microbiome-based metabolic therapies for ALD, and that the overall microbial community ecosystems.
Round 2
Reviewer 2 Report
This paper is only marginally better than the first iteration. There is too little information on how to leverage the micro biome to treat ALD. This is basically a summary of ALD.
Author Response
# Reviewer 2:
Revision 2: Comments and Suggestions for Authors
This paper is only marginally better than the first iteration. There is too little information on how to leverage the micro biome to treat ALD. This is basically a summary of ALD.
Response: We value the reviewers’ comments. Thank you for your valuable positive opinion about our manuscript.
We revised the Section number 5 (Probiotics and antioxidant activity in ALD) and Section number 7 (Summary and future outlook).
Nutritional care based on prebiotics and probiotics is an important part of ALD treatment. Probiotics are shown to be effective in reducing or preventing some ALD. Pro-biotics based on quantitative changes in bacterial overgrowth in the intestine, which is common in ALD patients. This probiotics of Akkermansia muciniphila and Lactobacillus spp., could help to improve the ALD survival rate. Recent studies investigating the use of pre-, pro-, and symbiotics in ALD and cirrhosis have found that they improve clinical and bio-chemical markers of liver disease. Probiotics provide promising new approaches to improve ALD.